Unfolding of α-helical 20-residue poly-glutamic acid analyzed by multiple runs of canonical molecular dynamics simulations

Ogasawara Naoki 1
http://orcid.org/0000-0003-0207-6271 Kasahara Kota 2 ktkshr@fc.ritsumei.ac.jp
Iwai Ryosuke 1
Takahashi Takuya 2
1 Graduate School of Life Sciences, Ritsumeikan University , Kusatsu, Shiga , Japan
2 College of Life Sciences, Ritsumeikan University , Kusatsu, Shiga , Japan
Salsbury Freddie Jr
Electronic publication date: 2018 May 15
Publication date: 2018
Volume: 6
Electronic Location ID: e4769
Received 2018 Feb 20; Accepted 2018 Apr 24
Copyright: © 2018 Ogasawara et al.
Copyright year: 2018
Copyright holder: Ogasawara et al.
License: This is an open access article distributed under the terms of the Creative Commons Attribution License, which permits unrestricted use, distribution, reproduction and adaptation in any medium and for any purpose provided that it is properly attributed. For attribution, the original author(s), title, publication source (PeerJ) and either DOI or URL of the article must be cited.
License URL: https://creativecommons.org/licenses/by/4.0/

Keywords: Molecular dynamics, Molecular simulation, Poly-glutamic acid, Conformational change, Peptide denaturation, Helix unfolding, Helix–coil equilibrium, Polypeptide, Helix–coil transition, Disorder

Funding: Japan Society for the Promotion of Science, Grant-in-Aid for Young Scientists JP16K18526 This work was supported by the Japan Society for the Promotion of Science, Grant-in-Aid for Young Scientists (Grant Number: JP16K18526). The funders had no role in study design, data collection and analysis, decision to publish, or preparation of the manuscript.

==============================
Elucidating the molecular mechanism of helix–coil transitions of short peptides is a long-standing conundrum in physical chemistry. Although the helix–coil transitions of poly-glutamic acid (PGA) have been extensively studied, the molecular details of its unfolding process still remain unclear. We performed all-atom canonical molecular dynamics simulations for a 20-residue PGA, over a total of 19 μs, in order to investigate its helix-unfolding processes in atomic resolution. Among the 28 simulations, starting with the α-helical conformation, all showed an unfolding process triggered by the unwinding of terminal residues, rather than by kinking and unwinding of the middle region of the chain. The helix–coil–helix conformation which is speculated by the previous experiments was not observed. Upon comparison between the N- and C-termini, the latter tended to be unstable and easily unfolded. While the probabilities of helix elongation were almost the same among the N-terminal, middle, and C-terminal regions of the chain, unwinding of the helix was enriched at the C-terminal region. The turn and 310-helix conformations were kinetic intermediates in the formation and deformation of α-helix, consistent with the previous computational studies for Ala-based peptides.

Introduction

Elucidation of the molecular mechanisms of protein folding is a central issue in physical chemistry. Since protein folding involves formation of secondary structural elements as building blocks of the tertiary structure (Richardson, 1981), understanding the dynamics of α-helical folding and unfolding, or helix–coil transition, is essential. The helix–coil transition has been extensively studied in both experimental and theoretical methods using mainly Ala-based polypeptides (Baldwin, 1995; Chen, Zhou & Ding, 2007; Neumaier et al., 2013) due to the high helix propensity of Ala residues (Spek et al., 1999). Another representative model peptide is poly-glutamic acid (PGA). Since the side-chain of Glu has a titratable group, the chemical nature of PGA can be modulated by the solution pH, and its helix–coil equilibrium can be controlled by pH adjustments (Nakamura & Wada, 1981; Clarke et al., 1999; Kimura et al., 2002; Inoue, Baden & Terazima, 2005; Causgrove & Dyer, 2006; Finke et al., 2007; Stanley & Strey, 2008; Donten & Hamm, 2013; Gooding et al., 2013). Previous experiments on the helix–coil transitions of PGA reported that compared to neutral environments, acidic environments enhance helix formation. The reported helix content of short PGAs in acidic environments varied from 0.3 to 0.6, whereas it is below the detectable limit in neutral pH (Clarke et al., 1999; Kimura et al., 2002; Finke et al., 2007). Detailed scenario of the dynamics of helix–coil transitions is still controversial. The previous reports have presented two different types of PGA conformations in acidic environments: (i) a single α-helix with denatured termini and (ii) multiple short α-helices connected by coil regions. Kimura et al. (2002) proposed that the single α-helical conformation arises via intermediate states with several short helices, based on Fourier-transform infra-red spectroscopy and circular dichroism (CD) experiments. Clarke et al. (1999) implied, based on stopped-flow CD measurements, that the single long α-helical conformation successively decomposes into multi-helical conformations. Finke et al. (2007) supported this scenario based on fluorescence resonance energy transfer (FRET) measurements.

In order to shed light on peptide conformational transitions at the atomic level, molecular dynamics (MD) simulation is a promising approach. This method has been applied to investigate the helix–coil transitions of Ala-based peptides, and the C-terminus has been reported to have a higher denaturing tendency compared to the N-terminus (Young & Brooks, 1996; Takano et al., 1999; Wu & Wang, 2001). In addition, the 310-helix and turn conformations were found to be kinetic intermediates for the helix–coil transitions (Young & Brooks, 1996; Takano et al., 1999). However, unlike that of the Ala-based peptides, helix-coil transitions of PGA peptides have not been studied using the all-atom MD method.

Here, we utilized the all-atom canonical MD method to simulate unfolding dynamics of a 20-residue PGA with fully protonated side chains, mimicking an acidic environment. Using the molecular model of a PGA with α-helical conformation as the initial structure, we repeated MD simulations for unfolding processes with different initial conditions. In total, 19-μs dynamics, consisting of three runs with 3.0 μs and 25 runs with 0.4 μs, were simulated. While various pathways of unfolding were observed in these 28 time courses, PGA unfolding was mainly seen to be triggered by denaturation of the termini, followed by propagation of the coil conformation toward the opposite side. Multiple-helix conformations implied by the previous experiments did not appear in the MD simulations.

Methods

Canonical MD simulations

Dynamics of a 20-residue PGA, in an explicitly solvated periodic boundary cell, was investigated by the canonical MD method. We prepared two α-helical PGA structures as the initial structures for simulation. The first was an α-helical structure, sampled from an ensemble, obtained by our replica-exchange MD (REMD) simulation, with an implicit solvent model. The details of the REMD simulation will be described elsewhere (R. Iwai et al., 2018, unpublished data). The second was an ideal α-helix, all the residues of which took the backbone dihedral angles φ = −60° and ψ = −45°, built using tLEaP software attached to AMBER package. The N- and C-termini of the PGA were capped with acetyl (Ace) and N-methyl (Nme) groups, respectively. All the carboxyl groups of the side-chains were protonated and the net charge of the PGA was zero. Each molecular model of the PGA was placed in the truncated octahedral cell and solvated by filling with TIP3P water molecules (Jorgensen et al., 1983). The number of atoms composing the molecular system with the simulated structure of PGA was 10,592, and that with the ideal α-helix was 11,081. After that, the energy minimizations were successively performed with the steepest descent and conjugate gradient methods; the number of steps was 250 for each. The systems were relaxed via a 200-ps NPT simulation using Berendsen barostat. For the system with the ideal helix, the heavy atoms in the PGA were constrained during the relaxation run. The final snapshots of these two systems, referred to as Sim and Ide, were used as the initial structures of the production runs (Fig. 1). Through the NPT relaxations, the cell dimensions shrank from 54.32 Å to 51.85 Å and from 55.10 Å to 52.66 Å for Sim and Ide, respectively. The convergence of cell volumes was confirmed in terms of the relative standard deviations in the last 100 ps of the NPT simulations (ca. 0.23%). As production runs, eight and 20 runs of simulations were performed with Sim and Ide systems, respectively. Accordingly, we termed these simulations as Sim1, Sim2, …, Sim8, and Ide1, Ide2, …, Ide20. The initial atomic velocities were randomly generated with different random seeds for each run. The simulation time of each run was 0.4 μs except for Sim1, Sim2, and Sim3 that lasted over 3.0 μs. These production runs were performed with the NVT ensemble at 300 K using the Langevin thermostat. The integration time step was 2.0 fs; the covalent-bond lengths and angles with hydrogen atoms were constrained with the SHAKE algorithm (Ryckaert, Ciccotti & Berendsen, 1977). The non-bonded pairwise potentials were truncated at 10 Å of the interatomic distance. For the potential energy calculations, AMBER ff99SB force field (Hornak et al., 2006) was applied. All the simulations were carried out using AMBER software.

Figure 1 The initial structures of MD simulations.

(A) The structure built by a REMD simulation, termed Sim. (B) The structure based on the ideal α-helix, termed Ide. (C) φ–ψ angles of second to 20th residues in Sim (triangles; the open triangles indicate the second and 20th) and Ide (circle; all residues have the same values).

Analyses

On the basis of the trajectories of the atomic coordinates, recorded every 20 ps in the simulations, the helix–coil transitions of a PGA were analyzed using DSSP software (Kabsch & Sander, 1983). DSSP recognizes the secondary structural elements in terms of hydrogen bonding patterns of the main-chains and categorizes them into the following eight classes: α-helix, 310-helix, π-helix, extended β-strand, isolated β-bridge, turn, bend, and others. Each class is represented by an alphabetical symbol; H, G, I, E, B, T, S, and O, respectively. Note that the symbol “O” is introduced in this paper for convenience, and it is denoted as “ ” (white or blank space) in the output of the DSSP software. The secondary structure content in the Ide trajectories was referred to as PIde (x; i) for the contents of the secondary structure x (any of the eight classes) at the i-th residue. The superscript “Ide” indicates that the ensemble was obtained from the 20 Ide runs with 0.4 μs each. The ensemble consisting of trajectories of 8 Sim runs with 0.4 μs each is indicated as the superscript “Sim”, and that of Sim1–Sim3 with 3.0 μs each is indicated as the superscript “Sim1–3”. The secondary structure content for the entire chain is presented as PIde (x). The transition probabilities of i-th residue, from the secondary structure x to y between the successive snapshots (20 ps of the time interval), PIde (y, x, i), were also evaluated. To measure the time required for the complete unfolding of an α-helix, we defined the unfolding time, tu, as the time corresponding to the first snapshot without α-helical residues in a trajectory.

Results

Micro-second dynamics of a PGA

In order to investigate long-term behavior of a PGA, we performed three runs of 3.0-μs MD simulations (Sim1, Sim2, and Sim3) with the same initial atomic coordinates but different atomic velocities (Fig. 1). The initially formed α-helix was deformed immediately after beginning the simulations in all the three runs (Fig. 2). The unfolding times, tu, defined as the time of the first snapshot without an α-helical residue in PGA for each trajectory, were 31.06 ns, 100.52 ns, and 7.38 ns in Sim1, Sim2, and Sim3 simulations, respectively. In the simulation with the longest unfolding time (Sim2), after unfolding of the initial α-helix, the helical conformation was temporarily reformed at the N-terminal half of the chain at around 0.2 μs (Fig. 2G). However, the reformed helix was unfolded at 0.34 μs, and a helix longer than 13 residues was not formed until the end. In the Sim1 simulation, although the initial helix was immediately unfolded, a long helix consisting of 17 residues was refolded and retained over a sub-micro second time scale (Fig. 2E). This helix was nucleated between 12th and 16th residues at 0.62 μs (Fig. 2D) and propagated over the range from second to 18th residues. While the N-terminal half of the helix was deformed at 0.84 μs (Fig. 2F), the latter half remained intact till 0.95 μs. On the other hand, re-formation of stable helix did not occur in Sim3, although several helix-nucleation events were observed. Overall, helix formation was a relatively rare event in this time scale. In addition, while several helix-nucleation events were observed, the nucleated helices disappeared immediately in most cases. Helix nucleation seemed to be coupled with the turn conformation (Figs. 2A–2C), the discussion on which will be taken up later. Formation of a β-sheet was also observed as a rare event. β-sheet formation in Sim2 was exceptionally stable and was retained during 0.63 μs (Fig. 2H).

Figure 2 The 3.0-μs time courses of the secondary structure elements and examples of snapshots for Sim1, Sim2, and Sim3 simulations (A, B, and C).

The time courses for Sim1, Sim2, and Sim3, respectively. The horizontal axis is the simulation time, and the vertical axis indicates the amino acid position in the peptide chain. Each block is filled by one of the eight types of colors regarding the secondary structure elements H, G, I, E, B, T, S, and O, and are indicated as red, maroon, dark-red, gray, black, dark-cyan, cyan, and white, respectively. (D, E, F, G, and H) Snapshots at (D) 0.625 μs in Sim1, (E) 0.804 μs in Sim1, (F) 0.842 μs in Sim1, (G) 0.199 μs in Sim2, and (H) 1.000 μs in Sim2.

In the time course of the secondary structural elements at each residue (Figs. 2A–2C), some “bands” could be observed; for example, the turn conformation was almost always formed at the 9th and 10th residues in Sim2. Since the tendency to form a turn at the 9th and 10th residues was not observed in the other runs, it is considered to be due to an initial condition, rather than an intrinsic propensity of the 9th and 10th residues. This indicates that there was the strong time-correlation of secondary structure formation, and 3.0 μs was not enough to reach an equilibrium state. The time course of the ensemble average of the helix content (summation over the α- and 310-helix conformations; P(H) + P(G)), for Sim1, Sim2, and Sim3 implies that the trajectories were not well-converged (Fig. 3). The gain of helix content in 0.5–1.0 μs of Sim1 corresponds to the refolding of the α-helix mentioned in the previous paragraph (Figs. 2A and 2D–2F). While the helix content of the three trajectories became converged to similar values with the evolution of time, they still acquired different values at the end of the simulations. The helix content in the full-length trajectories of Sim1, Sim2, and Sim3 were 0.14, 0.12, and 0.078, respectively. In addition, the time courses of the end-to-end distance and radius of gyration also showed slow equilibrations of the conformations (Fig. S1). These results imply that equilibration of the system requires longer time scales.

Figure 3 The time course of helix content averaged over accumulated time duration of each trajectory in Sim1, Sim2, and Sim3.

Unfolding dynamics

Non-equilibrium processes involved in the transformation of an α-helix into denatured structures were analyzed by scrutinizing the first part of each trajectory. We additionally performed 25 short (400 ns for each) simulations and analyzed the unfolding processes of the 28 simulations in total. Note that eight of them started from an α-helical conformation obtained from a simulation (Sim1–Sim8; Fig. 1), and the remaining 20 started from an artificially built ideal α-helix (Ide1–Ide20; Fig. 1). As a result, all the 28 runs showed corruption of the α-helical conformation within 400 ns (Figs. 4 and 5). The unfolding times (tu) varied from 7.38 ns (Sim3) to 380.70 ns (Sim6), and the average, median, and the standard deviation (SD) were 75.63 ns, 36.02 ns, and 92.18 ns, respectively (Table 1). There was no statistically significant difference between Sim and Ide simulation results; the average (median; SD) of tu were 72.65 ns (36.02 ns; 79.88 ns) and 83.09 ns (37.32 ns; 123.97 ns), for Sim and Ide, respectively.

Figure 4 The 400-ns time courses of the secondary structure elements of Ide1–20 for the panels (A)–(T), respectively.

See the legend of Figs. 2A–2C.

Figure 5 The 400-ns time courses of the secondary structure elements of Sim1–8 for the panels (A)–(H), respectively.

See the legend of Figs. 2A–2C.

Table 1 Unfolding properties of each run.

Run-ID	tu	Unfolding order	P(H) + P(G)	
Ide1	8.52	N,C,M	0.34	
Ide2	36.98	C,N,M	0.34	
Ide3	88.26	C,N,M	0.64	
Ide4	10.30	C,M,N	0.13	
Ide5	74.82	C,M,N	0.20	
Ide6	47.62	C,N,M	0.32	
Ide7	18.10	C,N,M	0.15	
Ide8	40.42	C,N,M	0.30	
Ide9	23.88	C,N,M	0.14	
Ide10	101.34	N,C,M	0.47	
Ide11	257.92	C,N,M	0.60	
Ide12	16.32	C,M,N	0.30	
Ide13	29.52	N,C,M	0.62	
Ide14	19.40	N,C,M	0.45	
Ide15	249.24	C,N,M	0.60	
Ide16	13.02	N,C,M	0.09	
Ide17	23.62	C,N,M	0.14	
Ide18	35.06	C,N,M	0.44	
Ide19	192.74	N,C,M	0.50	
Ide20	165.86	N,C,M	0.41	
Sim1	31.06	C,N,M	0.15	
Sim2	100.52	C,N,M	0.12	
Sim3	7.38	C,N,M	0.08	
Sim4	79.66	C,M,N	0.51	
Sim5	24.38	C,M,N	0.22	
Sim6	380.74	N,C,M	0.65	
Sim7	87.06	C,N,M	0.35	
Sim8	23.22	C,N,M	0.08	

The unfolding trajectories varied among the 28 trajectories. The fastest unfolding was observed in Sim3. The helix deformed from both the N- and C-termini immediately after the simulation began (Fig. 5C). As described above, while a single-turn helix sometimes formed at the N- and C-termini after unfolding, they did not grow into a longer helix. The bend conformations were stably formed at the fifth, sixth, seventh, 10th, and 11th residues during 400 ns. On the contrary, Sim6 showed the slowest dynamics of unfolding. While three or four residues from the N-terminus were immediately deformed, the remaining part of the helix was retained for a long time (Fig. 5F). As described above, strong time correlations were observed in all the trajectories (Figs. 4 and 5). After immediate unfolding of the α-helix, a denatured conformation of the peptide was not randomized in this time scale.

For all the 28 trajectories, unfolding mechanisms were analyzed in terms of the order of deformation for each region in the polypeptide chain. We classified the residues into three regions; i.e., the N-terminal region (second to seventh residues), the middle region (eighth to 13th residues), and the C-terminal region (14th–19th residues). The first and 20th residues were discarded because of the following reasons: they would be highly influenced by the truncation of the chain; the main-chain hydrogen bonding pattern of the first residue cannot be defined due to lack of the N-terminal neighbor; all the regions should have the same number of residues. Next, the order of unfolding, for these regions, was assessed based on the helix content of each region in the time period ranging from the beginning of simulation to the unfolding time, tu. As a result, the unfolding process beginning with the deformation of the middle region was not observed, and all the unfolding processes began with unwinding of one of the terminal regions (Table 1). In addition, coil regions propagated toward both the directions in many cases. There are two possible scenarios for completion of unfolding from any terminus: (i) the coil region appears in a terminus and elongates toward the opposite terminus (“N, M, C” and “C, M, N” in Table 1), and (ii) the opposite terminus is successively unfolded followed by elongation of coil regions from both the termini to the middle (“N, C, M” and “C, N, M” in Table 1). The fact that the former scenario was observed in only three and two runs among 20 Ide and 8 Sim runs, respectively, suggests the latter being the major way of α-helix unfolding in this system.

When comparing the N- and C-termini of the peptide chain, unfolding from the C-terminus was preferred over that from the N-terminus; 13 out of the 20 Ide runs and seven out of the eight Sim runs showed unfolding from the C-terminus. Difference between the two termini was clearer in Sim runs than in Ide runs, probably because of the slightly distorted initial structure of Sim (Fig. 1). The ensemble averages of residue-wise α-helix contents in Ide1–20 with 0.4 μs each (PIde (H; i)), Sim1–8 with 0.4 μs each (PSim (H; i)), and Sim1–3 with 3.0 μs each ((PSim1–3 (H; i)) also showed a lower helical tendency at the C-terminus than at the N-terminus (Fig. 6). The previous simulation studies (Young & Brooks, 1996; Wu & Wang, 2001; Finke et al., 2007) had also reported that helix formation of the C-terminal residues was unstable compared to that of the N-terminal ones.

Figure 6 Residue-wise secondary structure content of α-helix (H; solid gray line), 310-helix (G; dashed black line), and turn conformations (T; solid black line).

(A) The average over 20 Ide runs (PIde (x; i)). (B) The average over the 400-ns trajectories of eight Sim runs (PSim (x; i)). (C) The average over 3.0-μs trajectories of Sim1, Sim2, and Sim3 (PSim1–3(x; i)).

Secondary structural transitions

To analyze the detailed mechanisms of conformational transitions in shorter time scales, we assessed the probability of the event that the i-th residue in the secondary structure x at time t is transformed into y at time t + 20 ps; the averaged probability over the 20 Ide runs is referred to as PIde (y, x; i). For simplicity, we focused on the four classes of secondary structural elements; H, G, T, and HGT¯, which means any of the other five structural elements (I, E, S, B, and O). The cases i = 2, 11, and 19 were analyzed as representatives of the N-terminal, middle, and C-terminal residues, respectively (Table 2). The C-terminal residues showed a weaker tendency to retain the α-helical conformation compared to the other residues (PIde (H, H; 2) = 0.94, PIde (H, H; 11) = 0.96, and PIde (H, H; 19) = 0.58). The weaker tendency to retain the same conformation in the C-terminal region was also observed in the other secondary structures. The results of Sim runs were qualitatively consistent with that of Ide runs (Table S1).

Table 2 Probabilities of secondary structure transitions.

i	2	11	19	
x\y	H	G	T	HGT¯	H	G	T	HGT¯	H	G	T	HGT¯	
H	0.94	0.01	0.03	0.02	0.96	0.01	0.03	0.00	0.58	0.01	0.34	0.07	
G	0.08	0.60	0.22	0.10	0.15	0.55	0.28	0.02	0.04	0.50	0.32	0.14	
T	0.04	0.04	0.79	0.13	0.07	0.04	0.88	0.01	0.13	0.03	0.69	0.15	
HGT¯	0.00	0.00	0.02	0.98	0.00	0.00	0.04	0.95	0.00	0.00	0.03	0.97	

The helix–coil transitions mainly occurred via the turn conformation. More than half of the conformational transitions from the α-helix directed to the turn conformation; PIde(T,H;i)/PIde(H¯,H;i) for i = 2, 11, and 19 were 0.52, 0.73, and 0.80, respectively, where H¯ denotes the secondary structure other than H. In addition, formation of the α-helix via turn was enriched in the C-terminal residue; PIde(H,T;i)/PIde(T¯,T;i) for i = 2, 11, and 19 were 0.19, 0.62, and 0.42, respectively. Thus, the turn conformation can be considered as an intermediate state in the helix–coil transition, especially at the C-terminus. Another intermediate in the α-helix formation is the 310-helix. While a major destination state of a 310-helix was the turn (PIde(T,G;i)/PIde(G¯,G;i) for i = 2, 11, and 19 were 0.55, 0.62, and 0.64, respectively), it also transformed into an α-helix, especially at the middle position; PIde(H,G;i)/PIde(G¯,G;i) for i = 2, 11, and 19 were 0.20, 0.34, and 0.076, respectively (Fig. S2). This result agreed with the previous theoretical studies, which reported that the 310-helix is not a thermodynamic intermediate but could be a kinetic intermediate (Young & Brooks, 1996; Wu & Wang, 2001).

In addition to the position of amino acids in the polypeptide chain, effect of the α-helical ends was analyzed. We focused on segments consisting of three consecutive residues in the chain, and the state of the segment was defined as the combination of secondary structures of the three residues, grouped into the two classes, i.e., α-helix (“H”) and others (“−”; it has the same meaning as “H¯”). The state of a segment was divided into the following seven classes: “HHH”, “HH–”, “–HH”, “H–H”, “H– –”, “– –H”, and “– – –.” The state “–H–” is impossible, because α-helical conformation coincides with at least four consecutive residues. The probability of the event that the central residue of a segment forms an α-helix at the next snapshot (20 ps later) was analyzed for each class. For instance, probability for the class “HH–”, denoted as PIde (H, HH–), means the probability to retain α-helical conformation for the residue at the C-terminal end of an α-helix, regardless of the position in the chain (i). The probability of deformation of the C-terminal end of an α-helix can be shown as PIde (–, HH–) = 1−PIde (H, HH–). The probabilities are summarized in Table 3; the case of Sim runs is shown in Table S2. We found that a residue at the interior of an α-helix was more stable to maintain the α-helical conformation, compared to the terminal residues; PIde (H, HHH) = 0.97. It is noteworthy that the C-terminal end of an α-helix is more frequently deformed than the N-terminal one; PIde (H, HH–) = 0.74 and PIde (H, –HH) = 0.92. In addition, α-helix elongation toward the C-terminus was enriched compared to that toward the opposite direction; PIde (H, H– –) = 0.23 and PIde (H, – –H) = 0.04. The C-terminal end of an α-helix unstably changed its conformation while the N-terminal end tended to retain its conformation.

Table 3 Probabilities of helix folding and unfolding in Ide runs.

	All	N1	M2	C3	
PIde(H, HHH)	0.96	0.97	0.97	0.91	
PIde(–, HH–)	0.26	0.16	0.25	0.30	
PIde(–, –HH)	0.08	0.06	0.13	0.34	
PIde(H, H– –)	0.23	0.24	0.22	0.24	
PIde(H, – –H)	0.04	0.03	0.07	0.05	
PIde(H, H–H)	0.09	0.03	0.10	0.08	
PIde(H, – – –)	0.02	0.02	0.02	0.02	
Notes:

1 The N-terminal region consisting of the second to seventh residues.

2 The middle region consisting of the eighth to 13th residues.

3 The C-terminal region consisting of the 14th–19th residues.

We also evaluated the relationship between the two definitions of position; position in an α-helix (the N-terminal end, internal, and the C-terminal end) and position in the polypeptide chain (the N-terminal region [2 ≤ i ≤ 7], middle region [8 ≤ i ≤ 13], and C-terminal region [14 ≤ i ≤ 19]). The probability of helix–coil transitions in the center of a three-residue segment x was assessed for each of the three regions y: PIde (–, x; y) = 0.04, where x is “HH–” or “–HH” for the C- and N-terminal ends of an α-helix, respectively, and y is any of “N”, “M”, and “C”, for the N-terminal, middle, and C-terminal regions, respectively. The probabilities to unfold the N- and C-terminal ends of an α-helix varied with respect to the position of the ends in the entire chain; namely, higher probabilities were observed in the C-terminal region of the peptide chain (PIde (–, HH–; C) > PIde (–, HH–; N) and PIde (–, –HH; C) > PIde (–, –HH; N) in Table 3). While residue-wise α-helical content (Fig. 6) and α-helix retention probability (Table 2) indicate the highest α-helical propensity for the middle region, the lowest probabilities to unfold the ends of α-helix were found in the N-terminal region. In contrast, probabilities for elongation of an α-helix were almost the same for all the three regions (see PIde (H, H– –) and PIde (H, – –H) in Table 3). Therefore, an α-helical PGA tended to unfold from the C-terminus.

On the other hand, the α-helix nucleation was observed in low probabilities regardless of positions in the chain; PIde (H, – – –) = 0.02 for all three regions.

Discussion

In this study, we examined the dynamics of a 20-residue PGA with 28 runs of all-atom canonical MD simulations. While three of them simulated 3.0-μs time courses, the systems were not well-equilibrated (Fig. 3) and complete refolding of the α-helix was not observed (Figs. 4 and 5). The time scale required for α-helix formation by PGA, still remains controversial. The suggested time-scale varies from sub-micro to milliseconds (Clarke et al., 1999; Kimura et al., 2002; Causgrove & Dyer, 2006; Qin et al., 2014). Our simulation results imply that a time range of few micro-seconds is too short to refold PGA in acidic environments.

We mainly focused on the non-equilibrium dynamics of unfolding processes and repeated 28 runs of simulations with the two different initial α-helical structures. The results from these two initial structures were qualitatively similar. Higher stability of the α-helical conformation was shown to be in the middle of the polypeptide chain than at the termini. All the unfolding processes of the α-helix began from a terminus, but a helix–coil–helix conformation was not stably observed. In many cases, the unfolding proceeded toward both directions, rather than starting from a terminus and ending at the opposite. In addition, unfolding from the C-terminal side was preferred over that from the N-terminal side (Table 1). The probability of retention of α-helix at each residue was lower in the C-terminus than in the N-terminus (Table 2). While the probabilities of α-helix elongation were almost the same irrespective of whether the end was located at the N-terminus, middle, or C-terminus of the polypeptide chain, the probabilities of unwinding of the α-helix tended to be higher at the C-terminus of the chain (Table 3). The instability of α-helix at the C-terminus was due to the enhancement of unfolding, rather than reduction of folding. In the process of folding and unfolding of the α-helices, the turn and 310-helix conformations can be kinetic intermediates as consistent to the precedent studies (Young & Brooks, 1996; Wu & Wang, 2001; Pal, Chakrabarti & Basu, 2003).

Despite the wide acceptance of the all-atom MD method, there are still some issues under consideration. First, treatment of denatured proteins has not been fully validated in current force fields. Underestimation of the radius of gyration of denatured proteins by standard force fields and water models has been previously reported (Piana, Klepeis & Shaw, 2014). While there is no gold standard yet, some improved force fields and water models have been proposed to simulate denatured proteins (Piana et al., 2015; Henriques & Skepö, 2016; Huang et al., 2016). Second, although the force field applied here, AMBER ff99SB, is one of the standard force fields, there are some reports about its weakness; e.g., underestimation of helix stability (Sorin & Pande, 2005) and discrepancy with the quantum mechanical calculations (Takano, Kusaka & Nakamura, 2016). Third, finite-size effects have been reported for the helix-stability of a model polypeptide (Weber, Hünenberger & McCammon, 2000; Kastenholz & Hünenberger, 2004; Reif et al., 2009; Kasahara, Sakuraba & Fukuda, 2018). To avoid this problem, we used the large periodic boundary cells, which have at least a 10 Å margin between the solute termini and the cell boundaries, and the cell size was well equilibrated via the NPT simulations.

In fact, helix content in the simulated ensembles (Figs. 3 and 6) were lower than the experimentally reported values, which is in the range of 0.3–0.6. The ensemble averages [and SD] of end-to-end distances (19.11 [8.17], 19.57 [7.61], and 15.71 [5.85] Å for Ide, Sim, and Sim1–3, respectively) were inconsistent with the FRET measurements by Finke et al. (2007), which were 23–24 Å at pH 4. However, differences in the experimental method and conditions may cause differences in the helix content (Kimura et al., 2002), since precise measurement of the latter for short peptides is not straightforward (Kelly, Jess & Price, 2005; Greenfield, 2007). Discussion on the quantitative aspects of the results, e.g., helix contents and folding kinetics, provided by both the experimental and theoretical methods in this study, should be carefully considered. From qualitative aspects, our results were consistent with the reported theoretical studies, in spite of several differences in the materials and methods, e.g., peptide sequence, parameters, and sampling methods. For example, the weaker helix formation propensity at the C-terminus and the kinetic intermediates of helix–coil transitions were consistently concluded from this study in agreement with the previous theoretical studies. They are robust conclusions, regardless of adjustable settings and simulation methods. In addition to that, our simulation results provide statistics of kinetic details of helix–coil transition by multiple runs of canonical MD. The weaker helix formation propensity at the C-terminus is due to high frequency of unwinding rather than disfavoring of folding. Helix–coil–helix conformations speculated by previous experiments were not observed.

Note that effects of peptide length, which is one of the most important determinants of the helix–coil transitions of polypeptide, are not analyzed in this study. In general, the microscopic behavior of peptides depends on peptide length (Gómez-Sicilia et al., 2015). While this study focused only on the behavior of 20-residue PGA by following the previous study by Finke et al. (2007), some other previous experiments reported the effects of the length of PGA; for example, Clarke et al. examined 34-, 57-, and 163-residue PGAs (Clarke et al., 1999), Kimura et al. used 34- and 190-residue PGAs (Kimura et al., 2002), and Donten and Hamm used 20-, 50-, and 440-residue PGAs (Donten & Hamm, 2013). They demonstrated that longer PGAs tend to have slower folding kinetics and higher helix contents. For future works, simulating systems with longer PGAs would be useful for understanding the molecular mechanisms of effects of peptide length.

Conclusion

In this study, the unfolding mechanism of α-helix in 20-residue PGA was investigated using all-atom canonical MD simulations. Our results suggested that the unfolding was triggered by unwinding of a terminus, whereas the multiple short-helical conformations, implied in the previous experiments (Clarke et al., 1999; Kimura et al., 2002; Finke et al., 2007), were not stably observed in the simulated trajectories within the micro-second time-scale. The instability of C-terminus is consistent with the previously reported result from generalized ensemble simulations of the poly-Ala peptides (Young & Brooks, 1996; Takano et al., 1999; Wu & Wang, 2001). The mechanism of helix–coil transitions, shown here, might reflect the nature of the peptide backbone, and provide insight into the helix–coil transitions for general cases of polypeptides.

Supplemental Information

Supplemental Information 1 Table S1. Probabilities of secondary structure transitions.

Click here for additional data file.

Supplemental Information 2 Table S2. Probabilities of helix folding and unfolding in Sim runs.

Click here for additional data file.

Supplemental Information 3 Fig. S1. Time course of (A) end-to-end distance and (B) radius of gyration of PGA in Sim1 (red), Sim2 (green), and Sim3 (blue) runs.

Click here for additional data file.

Supplemental Information 4 Fig. S2. Probabilities of transition between different secondary structures in Ide runs.

The horizontal axis indicates the position of each residue, i. Transitions from α-helix (H), 310-helix (G), and turn (T) are shown in solid, dashed, and dotted lines, respectively. Destination states are indicated as red, maroon, dark-cyan, and green, for H, G, T, and structural elements other than these, respectively.

Click here for additional data file.

Supplemental Information 5 Trajectories of secondary structural elements.

The data includes the two initial structures and trajectories of secondary structural elements of all the residues in all the snapshots generated in this study. All the figures and tables in this manuscript were generated based on this raw data.

Click here for additional data file.

The supercomputer resources were provided by the HPCI System Research Projects (Project IDs: hp170020 and hp170025) and the National Institute of Genetics, Research Organization of Information and Systems, Japan. We thank Tomoya Hirano for help with data analyses.

Additional Information and Declarations

Competing Interests

Author Contributions

Data Availability

The authors declare that they have no competing interests.

Naoki Ogasawara performed the experiments, analyzed the data, contributed reagents/materials/analysis tools, prepared figures and/or tables, authored or reviewed drafts of the paper, approved the final draft.

Kota Kasahara conceived and designed the experiments, prepared figures and/or tables, authored or reviewed drafts of the paper, approved the final draft.

Ryosuke Iwai performed the experiments, contributed reagents/materials/analysis tools, authored or reviewed drafts of the paper, approved the final draft.

Takuya Takahashi conceived and designed the experiments, authored or reviewed drafts of the paper, approved the final draft.

The following information was supplied regarding data availability:

The raw data are provided as a Supplemental File.

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
