# Peer review of "Unfolding of α-helical 20-residue poly-glutamic acid analyzed by multiple runs of canonical molecular dynamics simulations"

_PeerJ, doi:10.7717/peerj.4769_

## Round 0.1 · original submission · Minor Revisions

Review two especially makes some important points. Though NVT vs NPT might not matter depending on the system size.

Reviewer 1 ·

Basic reporting

This is a tedious but useful study of the 20-residue poly-glutamic acid.
The poly-glutamic acid may come with various lengths and its properties depend on the length. There is no mention of this in the manuscript. One paper that should be referred to in this context is Gomez-Sicilia et al. PLoS Comp. Biol. 11: e1004541 (2015). The authors comment about the necessity of having force fields that would be more adequate for the IDP systems. There is already at least one such a force field available: Huang et al. Nature Methods 14: 71 (2017). At length 20, the system is also an IDP but these aspects are not characterized adequately. One way to do it is to provide values of the average end-to-end distance, its dispersion, and the average radius of gyration. The time-dependent structure has other interesting features than merely the secondary-structure content.

Experimental design

The theoretical set-up is appropriate.

Validity of the findings

The findings appear to be valid but limited.

Additional comments

Same as in point 1.

Reviewer 2 ·

Basic reporting

Clear and unambiguous, professional English used throughout.


Literature references, sufficient field background/context provided.


Professional article structure, figs, tables. Raw data shared.

Self-contained with relevant results to hypotheses.

Experimental design

Original primary research within Aims and Scope of the journal.

Research question well defined, relevant & meaningful. It is stated how research fills an identified knowledge gap.

Rigorous investigation performed to a high technical & ethical standard.

Methods described with sufficient detail & information to replicate.

Validity of the findings

Data is robust, statistically sound, & controlled.

Conclusion are well stated, linked to original research question & limited to supporting results.

Additional comments

The manuscript studies helical unfolding or helix/coil transition of a 20-residue Polyglutamic acid (PGA) in an acidic environment through explicit canonical (NVT) MD simulations to verify with two previously reported helical structures of PGA: single alpha-helix or short alpha helices separated by coil. The authors ran simulations with two different alpha helical conformations as starting structure – one with a helical conformation obtained from their previous MD study and the other, which they refer to as ideal alpha helix, was curated with all backbone phi and psi angles in the helical range. A total of 8 and 20 simulations with different starting velocities were run with the helical structure obtained from their MD study and with the ideal helical structure respectively. Among these, 3 simulations with the MD-obtained helix was run for 3 microseconds to account for long-term dynamics, the rest being all 400 ns each. The authors observed different unfolding times for the first three simulations, with transient helix formation after the first unfolding event, except for one of the simulations. The majority of simulations showed helix-coil transitions starting from both the termini and extending to the middle segment, and that the turn conformations were highly coupled with such transitions as obtained from their ensemble averaged transition probabilities starting from the ideal helix. They also obtained coupling of the 3-10 helices with alpha helices. They claimed that both the turns and 3-10 helices as kinetic intermediates in the helical unfolding pathway.
Overall, the manuscript reports well on conformational changes on a set of long simulation results from two different structures of a single protein, which is extensive. The secondary structure transition probability analyses, although ensemble averaged is satisfying. However, there are few things that may need to be clarified. Firstly, in the simulation method, they have used ensembles generated by NVT. There must be a good reason for this. In order to model conformational changes, that is helix-coil transitions in this case, it may be a good idea to use NPT as most experimental measures are obtained under constant pressure. This makes more sense in the light of their introduction to the topic, where the authors indicate towards verifying their findings with previously reported FTIR and CD spectroscopy. Secondly, in Figure 2, which shows the cumulative helix averages over the 3.0 microseconds runs (Sim1, Sim2 and Sim3) with same initial structure, but different starting velocities record quite different helical probabilities: especially with Sim 1 there is a sudden increase in helices between 0.5 and 1.5 microseconds. The authors need to give better explanation (rather than sampling time) why such a trend is seen with one simulation (may be some close side chain contacts stabilizing the helices?), but not the other two, although they have noted this in the text (lines 156-159). Finally, the claim in the conclusions and the abstract that turn and 3-10 helices are kinetic intermediates in the helix-coil transitions are based only on 3 (N-term, C-term and middle) out of a total of 20 residues in their analyses. In order to gain statistically significant numbers (and the kinetic intermediate claim), one has to consider the evolution/probabilities of at least 11 out 20 residues (>50%).

---

## Round 0.2 · accepted · Accept

The comments have been addressed, I am happy to Accept your submission.

# Reviewer 1 ·

Basic reporting

I am satisfied with the revised version of the manuscript.

Experimental design

Appropriate.

Validity of the findings

Appear to be valid.

Additional comments

The revised version is ready for publication.